# Bacteriophages as Biotechnological Tools

**DOI:** 10.3390/v15020349

**Published:** 2023-01-26

**Authors:** Mariana Alves Elois, Raphael da Silva, Giulia Von Tönnemann Pilati, David Rodríguez-Lázaro, Gislaine Fongaro

**Affiliations:** 1Laboratory of Applied Virology, Department of Microbiology, Immunology and Parasitology, Federal University of Santa Catarina, Florianópolis 88040-900, Brazil; 2Microbiology Division, Faculty of Sciences, University of Burgos, 09001 Burgos, Spain; 3Research Centre for Emerging Pathogens and Global Health, University of Burgos, 09001 Burgos, Spain

**Keywords:** bacteriophages, applications, biotechnological

## Abstract

Bacteriophages are ubiquitous organisms that can be specific to one or multiple strains of hosts, in addition to being the most abundant entities on the planet. It is estimated that they exceed ten times the total number of bacteria. They are classified as temperate, which means that phages can integrate their genome into the host genome, originating a prophage that replicates with the host cell and may confer immunity against infection by the same type of phage; and lytics, those with greater biotechnological interest and are viruses that lyse the host cell at the end of its reproductive cycle. When lysogenic, they are capable of disseminating bacterial antibiotic resistance genes through horizontal gene transfer. When professionally lytic—that is, obligately lytic and not recently descended from a temperate ancestor—they become allies in bacterial control in ecological imbalance scenarios; these viruses have a biofilm-reducing capacity. Phage therapy has also been advocated by the scientific community, given the uniqueness of issues related to the control of microorganisms and biofilm production when compared to other commonly used techniques. The advantages of using bacteriophages appear as a viable and promising alternative. This review will provide updates on the landscape of phage applications for the biocontrol of pathogens in industrial settings and healthcare.

## 1. Introduction

Bacteriophages, also called phages, are viruses that essentially infect bacteria and are the most abundant entities on earth. They were independently discovered in 1915 by Frederick Twort in England and in 1917 by Félix d’Hérelle in France [1]. Since then, it has contributed to a broad array of developments in molecular tools. Alfred Hershey and Martha Chase evidenced that genes were composed of DNA using phages in an experiment named “Waring blender experiment” [2]. The phages also contributed to discoveries related to genetic control of enzymes, virus synthesis, and mutagenesis of bacterial genes for functional studies [3,4,5], to mention a few.

Phages are obligate intracellular parasites, consisting of a nucleic acid surrounded by a three-dimensional structure, the capsid, which may or may not be surrounded by a lipoprotein envelope. They may have a tail, a spiral contact sheath, where there are usually fibers that present the proteins responsible for binding to the receptor, that is, to the membrane of the target host. Most phages have the dsDNA (double-stranded) genome, but they can also have dsRNA, ssDNA (single-stranded), and ssRNA genomes [6,7,8].

For a long time, the taxonomic classification of bacteriophages was based on their genome type, morphology, and host range. In 1937, Sir Macfarlane Burnet showed that phages are different in size and resistance against physicochemical agents [9]. In addition, Holmes Ruska proved that phages were morphologically diverse and proposed a classification of viruses by electron microscopy in 1943 [10]. The Holmes classification constituted the phages into three families based on the host range and symptoms of the disease [11].

In 1962, Lwoff, Horne, and Tournier proposed virus classification based on the virion, its nucleic acid, and a Latinized nomenclature that included several phages [12].

In 1967, David Bradley used electron microscopy and acridine orange to classify tailed bacteriophages into phages with contractile tails, long noncontractile tails, and short noncontractile tails. In 1971, this system was officially adopted by the International Committee on Nomenclature of Viruses (ICNV) and enhanced with the names *Myoviridae*, *Styloviridae*, and *Pedoviridae* for the three morphotypes by Hans-Wolfgang Ackermann and Abraham Eisenstark of the Bacterial Virus Subcommittee in 1975 [13,14,15,16].

The family phages’ names, *Myoviridae* and *Podoviridae*, as known currently, were accepted by the International Committee on Taxonomy of Viruses (ICTV) in 1981 and *Siphoviridae* in 1984. Following the phage taxonomy story, the order *Caudovirales* (comprising *Myoviridae*, *Siphoviridae*, *Podoviridae*, *Ackermannviridae*, and *Herelleviridae* families), unifying all tailed phages, was proposed by Hans-Wolfgang Ackermann and approved in 1988 [17].

In the same way, based on the morphology, other phage families were classified as *Inoviridae* (filamentous capsid), *Microviridae*, *Tectiviridae*, *Corticoviridae*, *Leviviridae*, and *Cystoviridae* (polyhedral capsid), and *Plasmaviridae* (pleomorphic capsid) in 1978 [17].

However, with the advent of genomics, the sequencing of phage genomes revealed a higher genomic diversity, from the human gut to the deep ocean [18]. In addition, a genome organization-based taxonomy included new subfamilies and genera and virus relatedness based upon nucleotide or more common protein sequences or proteome comparisons. Based on that, molecular analysis can provide a robust classification guide for phages, even for those with highly divergent genome sequences and organizations, high rates of horizontal gene transfer, and mosaicism [19,20,21,22,23].

According to the National Center for Biotechnology Information (NCBI), in 2019, there were 8,437 complete bacteriophage genomes divided into 12 families and an unclassified group, with more than half being of the *Siphoviridae* family [18].

As mentioned earlier, phages are part of a complex ecosystem and are found in different environments, from the ocean to the microbial environment within a microorganism [1]. In marine environments, the bacteriophages play an important role due to their abundance and diversity, modulating microbial communities, generating genetic diversity, and nutrient cycling through bacterial mortality. In addition to morphological diversity, phages vary their replicative cycle according to the season. For example, in the western Antarctic Peninsula and the Canadian Arctic Shelf, prophages dominate in the spring while lytic infections prevail in the summer [18,24].

The diversity and abundance of phages in soils are directly dependent on the biome. It varies with soil type, chemical characteristics, and bacterial abundance. For viral abundance, a recent study indicates that these numbers can vary greatly, from 10³ to 10^9^ viral-like particles per gram of soil, the former referring to desert soils and the second to forest soils [18,25]. The human gut is another well-studied phage environment; estimates from human stool show a high abundance of these viruses, up to 10^8^ viral-like particles per milliliter of the sample. The *Caudovirales* order is the most abundant group of viruses found in the human gut, as an electron microscopy study has shown [26]. The intestinal phage community is stable over time, but rapid changes are observed early in life. The human virome is directly related to the organism’s health [18].

## 2. Life Cycles of Bacteriophages

In terms of the characterization of phage life cycles, there are strategies for infection and release. Related to encapsidation and location, phages can be: (i) intracellularly and unencapsidated. This state can be subdivided as a “Vegetative phase” as described by Lwoff in 1953, or productive cycle versus a prophage; (ii) intracellularly and packaged within mature virions, in other words, can be distinguished from phage genomes and therefore are not packaged until the virion release step; and (iii) intracellularly and encapsidated. In this case, free phages are no longer found within their bacterial host [27,28].

Based on these strategies of infection and release, phages can be lytic and non-temperate, in other words, when lytic phages do not display lysogenic cycles, or chronic and non-temperate, that is, when chronically released phages do not display lysogenic cycles. In the first strategy, the phage passes through a vegetative phase, and its genome is packaged into mature virions before the release of free phages. The second case is similar to the previous one—the phage passes through a vegetative phase—however, its genome is packaged into mature virions during the release of free phages [28].

The lytic and non-temperate phages are of great interest for the biological control of bacteria, as they can lysate the host cell, and the resulting progeny continues the cycle [29].

The next two strategies of infection and release of a phage include lytic and temperate phages, in other words, lytic phages that can display lysogenic cycles, or chronic and temperate phages, that is, chronically released phages that can display lysogenic cycles. In this scenario, in the first strategy, the phage can display a vegetative or prophage phase, and its genome is packaged into mature virions before the release of free phages. Additionally, similar to this, in the second strategy, the phage can display a vegetative or prophage phase; however, its genome is packaged into mature virions during the release of free phages [28].

The first step of a phage’s life cycle consists of the virion’s encounter with the bacteria, a process also called adsorption. In the adsorption process, the virion movement toward the bacteria can be differentiated into diffusion and non-diffuse movement (relative, turbulent, or bulk) [30]. The diffusion movement occurs when the phages are not attached to or entangled with materials. In addition, this movement can be influenced by different factors, such as particle size and morphology [31,32].

The non-diffuse movement is related to phage binding to nonhost materials, which can result in virion retention within fluids along with virion movement in association with ongoing currents. The non-diffuse movement can increase the likelihood of a phage encounter with a target cell if virions are moving in such a flow relative to the target bacteria. If the movement is a turbulent flow (e.g., microenvironments mixed), it can instead result in a faster virion movement relative to bacteria [33]. Lastly, virions also may be transported over longer distances, such as through the air, associated with dust, splashed water, or animals, serving as mechanical vectors [34,35,36]. However, it is important to note that this bulk movement is not necessarily relative to the positions of co-located bacteria. Therefore, it may not contribute to phage encounters with susceptible bacteria [30].

The following step of the adsorption process consists of the phage’s reversible attachment to the host bacteria. First, the attachment is reversible because there are no permanent changes in virion morphology. Then, the final step of the adsorption process culminates in a free virion irreversible attachment to a bacterial cell. The irreversible attachment occurs thanks to the binding of the secondary attachment proteins to the secondary receptor, which is stronger than the binding of the primary attachment protein to the primary receptor in the reversible attachment [30].

The next step of a phage’s life cycle consists of phage acquisition of bacteria, that is, the conversion of a free virion (seeds or spores for multicellular organisms) to a virocell (“living form” of the virus) [37]. In this step, the virion genome is translocated to the bacteria’s cytoplasm. To translocate their genomes, phages have to surpass the bacterial surface structures (e.g., glycocalyx, S-layers, and peptidoglycan cell walls). For that, most phages use their enzymes targeting these structures and introduce their nucleic acid into the host cytoplasm via a combination of mechanical and enzymatic action that results in a process known as genome injection [38,39,40].

In the bacterial cytoplasm, the phage genes are expressed, and its genome is replicated, starting virion morphogenesis. The next step comprises a phage virion’s accumulation intracellularly, at a constant rate, until phage-induced bacterial lysis. In some cases, loss of infection viability or cessation of phage production for chronically infecting phages can occur, as discussed before [41].

The movement, attachment, and genome translocation of the phage to the host bacteria occur similarly to what was described before with chronic and temperate phages or lytic and temperate phages displaying lysogenic cycles. However, in the lysogenic cycle, the phage’s nucleic acid recombines with the bacteria’s nucleic acid, forming a prophage, which replicates with the host chromosome and is transferred vertically from the initially infected cell to its progeny through cell division. The lysogenic cycle may provide immunity against infection by the same type of phage. Moreover, stress conditions such as ultraviolet light, mutagenic chemicals, or DNA damage can induce a shift to the lytic cycle [42].

## 3. Mechanisms and Determinants of Host Range

Several features make it possible for a bacteriophage to infect a host cell. One of the most well-studied is the mechanism of phage-host adsorption interaction, which is a primary determinant for phage specificity and is mediated by the viral receptor binding protein (RBP). RBPs differ from each phage by the type of phage morphology and cannot be recognized just by sequence homology with other known RBPs. These proteins, also known as spikes or fibers, are called tail fiber or tail spike proteins in tailed phages [43,44]. Furthermore, these proteins bind to polysaccharides (LSP, capsular polysaccharides, teichoic acids, etc.) or proteins (e.g., protein porin OmpC, FhuA) on the bacteria’s surface [43,45].

Adsorption is a process that occurs in two steps. The first step is the reversible binding of the bacteriophage RBPs with the receptors on the host’s surface. This process, as the name suggests, can be undone, and the specificity of this process varies between different groups of phages. The second step is irreversible binding, which comes before the genetic material’s insertion inside the host cytoplasm [46,47].

*Enterobacteria* phage T4 is one of the most well-studied prokaryotic viruses. Additionally, many fundamental concepts of molecular biology were discovered thanks to work with this phage. There is even a subfamily of T-like viruses, the *Tevenvirinae*, which belongs to the family *Myoviridae*, a group of tailed viruses with a long, contractile, and straight tail. The T4 phage infects *Escherichia coli* and related *Shigella* species. By adsorbing to the host cell, the tail fiber proteins must change their conformation. So, they can interact reversibly with the receptors on the bacterial surface. To attach to the host cell, T4 uses its protein gp37. For an irreversible binding, it is necessary that the gp12 protein, a short-tail fiber, interact with heptose residues of the *Escherichia coli* lipopolysaccharide (LPS) of the cell wall. To complete invasion, sheath-induced tail contraction injects the viral genome into the cytoplasm of the bacteria [48,49].

The mechanisms of phage-host initial interaction in tailless bacteriophages are still unclear. The tail exercises several functions, from host recognition to piercing cell walls for penetration. For phages that do not have them, other proteins must exercise these functions to infect bacteria [50]. *Escherichia* virus φX174 is a single-stranded DNA tailless phage. It has a cubic shape, and, as the name says, it infects *Escherichia coli.* φX174 was the first DNA genome to be sequenced; Fred Sanger and his team completed the work in 1977 [51]. This bacteriophage recognizes sugar residues of the LPS present in the cell wall of Gram-negative bacteria. The capsid of the φX174 has 60 F proteins with 12 spikes on the vertices with 5 G proteins each. One of the 12 spikes initiates contact with the host cell, then dissociates from the capsid. After this process, a series of conformational changes occur in the recognition proteins, including the de F protein of the phage, to stabilize the attachment [50].

Bacteriophages can alter their host range through modifications in the RBPs. Some phages can alter their RBPs over the generation with a mechanism based on the enzyme reverse transcriptase. The temperate *Bordetella* phage BPP-1 is an example of a phage that generates diversity in a gene, designated *mtd* (major tropism determinant), and specifies different tropisms for receptor molecules by a reverse transcriptase–mediated process. The reverse transcriptase enzyme introduces nucleotide substitutions at defined locations within *mtd* that result in tropism switching. Based on that, a huge repertoire of lig-and-receptor interactions is generated. Briefly, in this study, searching to understand the tropism presented by the BvgAS signal transduction system that controls the infectious cycles of Bordetella subspecies, the researchers found a region of variability, VR1, that differed between tropic variants. Located downstream from *mtd* is a second template repeat (TR) that never varied when sequences of phage with similar or different tropisms were compared. Adjacent to TR is a locus, called *brt* (*Bordetella* reverse transcriptase), which encodes an enzymatically active reverse transcriptase (RT) similar to the RT domains of group II intron maturases, bacterial retrons, and retroviral reverse transcriptases. Based on these, the researchers constructed a series of in-frame deletion and substitution mutations to determine the roles of VR1, the TR element, and the *brt* locus in phage infectivity and tropism switching. Deletion mutations in the *brt* loci of two variants resulted in fully infective phages that had completely lost the ability to switch tropism. Altering the conserved reverse transcriptase motif eliminated *brt* activity in vitro and tropism switching in vivo. Thus, they concluded tropism switching is a reverse transcriptase–mediated event [52].

Other groups of phages present multiple RBPs, so these phages can modify their host range by using a specific protein. Random mutation events on the RBP genes can also change phage specificity. The bacteriophage phi92 represents this mechanism. It encodes at least five different types of RBPs for different host ranges, which enables a recognition of encapsulated and nonencapsulated bacteria, including both *Escherichia coli* and *Salmonella enterica* species [45,53].

The RBPs can be engineered to manipulate the infection spectrum, producing phages with a customized host range; several works have already made this achievement [54,55,56,57]. A narrowed host-spectrum phage of *Streptococcus thermophilus* was created by homologous recombination of genes from two different phages [56]. Similarly, replacing the gene responsible for the determination of the host range (gp37) from one *Enterobacteria* phage with the gene of another phage was capable of expanding the host spectrum of infection of the first one. This recombination process gave the chimera phage the ability to infect other eight host bacteria besides the combined host range of the parental phages [55].

By modulating the phage host range, it is possible to refine the application of phages in individualized therapies. Therefore, circumventing the limitation of many already isolated phages to have a limited host range, customizing phages for specific infections, and saving the patient’s microbiota are possible in the case of a clinical application [55,58].

The immunity systems of prokaryotes also rule the host range of bacteriophages. RBPs are important in the host-phage initial interaction because, after the virus invasion of the host cell, the immune mechanisms start to act. Superinfection immunity, modification-restriction, and CRISPRs are some of the most studied mechanisms [59]. Superinfection immunity is a mechanism that controls a superinfection event, in which a second infection occurs in a host that is already infected and has a prophage in its genome [60]. Superinfection inhibition can occur by interfering with the invasion or replication of the superinfection virus. In phage λ, a temperate virus, resistance against superinfection happens through binding the virulence repressor cl to the oLoR operator. This operator covers up promoter genes that can conduct viral lysis and the replicative cycle [60,61].

Modification-restriction systems are also immunity mechanisms found in bacteria to resist phage infection. These systems act primarily by destroying foreign DNA before it initiates any replicative cycle. There are four types of these systems, generally based on two enzymes: restriction endonucleases and modification methyltransferases. The first one catalyzes the cleavage of the DNA, and the latter performs the methylation, or in other words, the addition of methyl groups in one nucleotide on each strand to prevent the degradation of its genetic material, and for the endonuclease performs its function. Some phages encode anti-restriction systems, which are systems to allow evasion of the restriction immunity, alteration, and occlusion of the restriction site are examples of how these mechanisms can act [59,62].

The CRISPR is a system of genetic elements that confers immunity for the bacteria to exogenous genetic elements, such as viruses and plasmids. CRISPR works as an adaptive immune system in a way that incorporates invasive sequences to degrade new invaders that have previously embedded sequences. It contains short sequences of bacterial genetic material that repeat themselves and are regularly separated by non-coding sequences. The non-coding sequences, called spacers, are acquired by a phage infection; e.g., the acquisition of a new spacer, called adaptation, is the first step of the process, following the CRISPR RNA (crRNA) creation. Finally, the cleaving of invading sequences processes are catalyzed by Cas (CRISPR associated) nucleases and guided by crRNAs. The Cas nucleases are encoded with spacers and repeat sequences [63,64,65].

## 4. Biotechnological Applications

The fundamental principle that makes phages versatile for several applications is their capacity to infect and kill specific pathogens. Other phage intrinsic characteristics, such as its coevolution with its host, its enzymes capable of lysis of the bacterial cell wall, and its integration into the host genome, also make them candidates for biotechnological applications (Figure 1) [66,67].

The application of bacteriophages is not limited to medical use in animals and humans. In agriculture, they are used to treat different problems with bacterial contamination [68,69,70]. In the food industry, they are used as a microbiological control method for undesirable pathogens, increasing food safety [71,72]. In addition, the proteins encoded by phages, such as endolysins (which degrade peptidoglycan) and holins (which break the cell membrane of the infected cell), have shown important antibacterial actions [73].

The use of phages for the treatment of bacterial infections is called phage therapy. Phage therapy is a method to combat bacterial infections using bacteriophages and has been a major focus of attention in recent years. This approach mainly uses viruses that are professionally lytic, so that they kill the pathogen without necessarily promoting any changes in its genome. The advantages of bacteriophage use include: (i) its specificity for target bacteria in the event of clinical application, which may considerably reduce the damage to the patient’s intestinal microbiota; (ii) self-limiting growth, meaning that they require their hosts to be constantly growing; (iii) if the bacterial pathogens for which they are specific are absent, they will not persist long enough; and lastly, (iv) replication at the site of infection.

Phage therapy can only be applied if the host range is well characterized and limited to the pathogen of interest. This feature makes it possible to use phages for the treatment of infections by themselves or in combination with antibiotics, allowing the reduction of chemical use. Over time, mostly because of the discovery of antibiotics, the phages were left aside. However, in recent decades, the research became focused on phages as a possible tool for bacterial treatment because of the rise in multidrug-resistant bacteria around the world and the decline in production and discovery of new antibacterial chemicals [74].

Phages have an advantage over antibiotics when it comes to target specificity. They are found in nature with different ranges of the host spectrum. Some can infect only specific strains of certain species of bacteria; therefore, finding a customized phage for each infection may be difficult. To this day, there is no licensed (by the US Food and Drug Administration or the European Medicine Agency) phage preparation for human purposes [74,75].

### 4.1. Human Phage Therapy: Advantages and Limitations

Several works show the positive effect of phage therapy on bacterial infection in humans. These are focused mainly on topical infection treatment of lung infection and otitis, with one report about cholera that has not yet been studied in a controlled clinical trial [76]. The species most present in these studies are Pseudomonas aeruginosa, *Staphylococcus aureus*, and *Achromobacter xylosoxidans*. The beneficial effects of the application of phages for these infections have been reported. In some of these studies, phages were the only antimicrobial treatment applied, and yet positive results were demonstrated [76,77,78,79,80,81,82,83]. Beneficial effects are not observed in every clinical trial compared to standard treatment. A study tested a cocktail composed of 12 bacteriophages to treat burn wounds infected with *Pseudomonas aeruginosa*. The results showed almost no difference between the treatments [80].

Another tool for antibacterial purposes is phage tail-like bacteriocins (PTLBs) or tailoring. These structures resemble the tail of bacteriophages, but without a head and the absence of a genome. As with phages, PTLBs also depend on RBPs to target a host bacterium. Tailocins, different from phages, kill the prokaryotic cell using the mechanism of membrane depolarization by making a pore in the membrane, allowing the passage of ions to the interior of the cell. Unlike phages that have a complete cycle going into the interior of the host cell, PTLBs only act on the cell surface, so for the bacteria to acquire resistance against it, the process must occur in the cell receptors only [58].

For a long time, it was thought that bacteriophages could not affect eukaryotic cells. However, recent studies have provided new insights about the interactions between them.

Bacteriophages can interact with the surfaces of mucus, for example. An in vitro experiment using a T4 bacteriophage and mucus-producing tissue cell line showed a binding between the bacteriophages and glycan residues displayed on mucin glycoproteins via Ig-like domains present in hoc capsid protein [84]. Since the attached bacteriophages have a higher probability of encountering the host bacteria than free virions, it is proposed that mucus adherence may help shape the gastrointestinal microbiome and prevent pathogens from colonizing the system [84,85,86].

The bacteriophages may also interact with cells of the immune system. A study assessed the effects of the bacteriophage ES2 on the expression of surface proteins CD86, CD40, and MHCII; the production of pro-inflammatory cytokines IL-6, IL-1α, IL-1β, and TNF-α by dendritic cells; and the activation of the NF-κB signaling pathway. The results showed that ES2 increased the expression of surface proteins and pro-inflammatory cytokines. In addition, the study also showed the activation and translocation of NF-κBp65 to the nucleus, leading to the activation of NF-κB signaling [87]. In the same way, another study observed the activation status of mammalian macrophages and TNF-α levels by two bacteriophages of *Escherichia coli*.

Cells of the immune system, such as those cited above, express pattern recognition receptors. These receptors include a family of receptors called Toll-like receptors that are capable of recognizing bacteriophage nucleic acids in endosomes.

In addition to the immune system, there are studies presenting interactions of bacteriophages with the respiratory system [88,89], central nervous system [90], gastrointestinal tract [86], urinary tract [91,92], and cancer [93,94,95], to cite some.

Among the disadvantages of phage therapy is the release of bacterial toxins (e.g., endotoxins) when the bacteria are lysed, which can worsen bacterial infection [96,97]. Furthermore, foreign proteins carried by bacteriophages may induce an overreaction or cause an imbalance in the immune systems of humans or animals [98,99].

Another disadvantage is the limited knowledge about phage diversity, lack of familiarity with basic safety issues for phage therapy application and proper phage selection [74,100,101,102].

Related to bacteriophage cocktails, this approach arises due to the high specificity of phages, which often leads to limitations in identifying the fitting strain. In emergency cases, for example, where there is a search for a corresponding strain of phage before treatment, the use of phage cocktails can increase the efficiency of pairing by increasing the range of action [103,104]. Phage cocktails can also delay the emergence of phage-resistant bacteria since multiple phages are interacting with bacteria. In addition, different strains of phages can complement one another by providing the necessary antimicrobial elements that one may be short of [103].

If the phage isolation is successful, genomic characterization must be carried out to guarantee the safety of the preparation. Therefore, it must be searched for integrase genes, antibiotic-resistant genes, or genes for bacterial virulence. If any of these genes are found, this phage no longer becomes a promising candidate to compose a cocktail. The difficulty to prepare formulations that keep phages stable, target specific tissues for the treatment of infections, and disperse bacterial residues formed after cell lysis are all problems that must be circumvented [105,106].

#### Horizontal Gene Transfer (HGT) in Prokaryotes

There are three mechanisms of horizontal gene transfer (HGT) in prokaryotes: transformation, conjugation, and transduction [107]. Bacteriophages mediate HGT through transduction, where the DNA from a donor bacterial cell is transferred to a recipient bacterial cell. Transduction can be specialized or generalized. The first one can occur only with temperate phages and is generated from the imprecise excise of adjacent host genes together with the phage genome. Therefore, the host DNA becomes part of the phage genome, is replicated, and all virions produced by the cell will carry the fragment of the host DNA [42].

Bacteriophage lambda is a classic example of a specialized transducing phage since it integrates into the genome of *Escherichia coli*, transfers genes related to the galactose metabolism and biotin biosynthesis, and confers a fitness advantage in certain environments.

The generalized transduction in turn can occur by both the lytic or lysogenic cycle and consist of random parts of the host DNA packed in bacteriophages. Briefly, the terminase complex recognizes a *pac* site homolog in the host chromosome and introduces a double-stranded break nearby. When the small subunit of the terminase complex recognizes the *pac* sequence, the large subunit starts to package the DNA into the procapsid until the capacity of the phage head is reached and a second double-stranded break is introduced. The filled virion passes through the tail maturation and detaches from the terminase complex. Then, the terminase complex attaches to the free end of the bacterial chromosome and encapsidates phage-sized fragments of the host genome until it disengages from the chromosome [42].

### 4.2. Applications of Phages in Livestock and Food Industry

Since the administration of phages directly to live animals until to ready-to-eat food, the biotechnological applications achieved using bacteriophages are diverse and still growing. Beyond the applications targeted in the food industry, bacteriophages are also widely studied and applied in areas such as health and sanitary treatment [108,109,110,111,112,113,114].

Among the applications in the most diverse scenarios, as explained above, different methodologies and approaches are employed. The search for optimization and improvement of these phage biotechnological processes is constant.

#### 4.2.1. Applications in Live Animals (Animals Phage Therapy)

Food safety begins on farms and in environments where animal husbandry is applied. However, large numbers of animals are housed in these environments, and often, these are kept confined, promoting the proliferation of agents related to infectious diseases among herds. In this scenario, animals act as reservoirs for different zoonotic bacterial pathogens that enter the food chain, and result in human diseases and death [115].

From this perspective, bacteriophages become a viable alternative for pathogen biocontrol that may be associated with animals. For this, there are methodologies such as bacteriophage suspensions inoculated directly on an animal’s food and water or indirect application in the immediate animal’s environment, where absorption is facilitated [116]. Phage delivery can be carried out orally when added to animal feed or water. Clavijo et al., 2019 evaluated the efficacy of SalmoFREE^®^, a patented phage preparation, which is effective against *Salmonella*. The product was applied to broilers, at different times, directly in the water. The presence of *Salmonella* was determined before and after treatment by a cloacal smear. The researchers observed that the product managed to control the bacterial population, with counts falling to 0% on day 34 compared to the control group [117].

Similar to the previous strategy, intragastric phage delivery can also be carried out, consisting of the addition of bicarbonate or another active agent for neutralizing stomach acid, followed by oral consumption of bacteriophage suspension, where some phages survive passage to the intestine. Ma et al., 2012 encapsulated bacteriophage K, effective against *Staphylococcus aureus*, in alginate microspheres with calcium carbonate to ensure the survival of the phage in the acidic environment of the stomach. Unencapsulated phages were completely deactivated, while encapsulated phages resulted in a 0.17 log10 bacteria reduction in two hours. Although the bacterial reduction is low, bacteriophage delivery using this strategy may be possible through adaptations of the delivery vehicle [118].

There are also related transdermal, epidermal, subcutaneous, epicutaneous, and intramuscular phage injections. Beyond blood or vascular delivery, in which the bacteriophage suspension is adjusted to be physiologically similar to the host’s blood, bypassing the harmful effect of the host’s stomach acid and providing an accurate viral titer [116].

#### 4.2.2. Pathogen Biocontrol in Food

In addition to the strategies using phages applied directly to live animals, others can also be applied to meat, dairy products, processed products, vegetables, fruits, and food packaging [51,119,120,121,122,123,124].

The phage applications in food are based on professional lytic phages, capable of infecting host bacterial cells, lysing them, and consequently eliminating the population of bacteria present in the food [125].

When used to control and eliminate bacterial populations in foods, bacteriophages should be selected carefully. Phages must have a professional lytic cycle, since lysogenic phages do not necessarily lyse the bacterial cell and can even transmit undesirable genetic elements [126]. Other relevant factors are that the effectiveness of phages varies between food matrices and that each bacterial serovar can show different degrees of susceptibility to phages. Thus, the identification and characterization of bacteriophages in different food matrices is very important before their application in biocontrol actions [127].

Several studies demonstrate the efficacy of bacteriophages in foods for the control of foodborne bacteria, such as *Listeria monocytogenes*, *Salmonella*, *Campylobacter*, *Escherichia coli*, *Clostridium,* and *Staphylococcus aureus*. The interest in these bacteria comes from the fact that they are most responsible for illnesses and deaths in the world, generating high costs and great concern for food companies and health care systems [128].

Currently, there are already several products containing phages approved by the United States Department of Agriculture (USDA) and marketed for the biocontrol of pathogenic bacteria. For the control of *Salmonella*, formulations are available, such as SalmoFresh^®^, a preparation with specific phages against *Salmonella* that is effective to treat foods that present a high risk of contamination and capable of killing the most common serotypes such as Typhimurium, Enteritidis, Heidelberg, Newport, Hadar, Kentucky, Thompson, Georgia, Agona, Grampian, Senftenberg, Alachua, Children’s, Reading, and Schwarzengrund. SalmoFREE^®^ product is a cocktail containing six different phages, effective against *Salmonella* spp. [129]. Studies have already shown the effectiveness of this product on fresh fruit, cucumbers, chicken, and turkey breast cutlets [130,131,132].

Aiming for the control of *Escherichia coli*, EcoShield^®^, a phage preparation effective against *Escherichia coli* and Shiga toxin-producing *Escherichia coli* (STEC), is available on the market [133]. In addition, there are *Listeria*-targeted formulations such as ListShield^®^, a phage cocktail formulation specific against *Listeria monocytogenes* that can be applied to fresh produce, dairy products, fish, and ready-to-eat foods [134]. Additionally, ListexTM, a P100 phage suspension formulation, is also effective against *Listeria monocytogenes* for application in meat, fish, seafood, dairy products, and other ready-to-eat foods, in a concentration of up to 1 × 10^9^ Plaque-Forming Units (PFU) per cm^2^ [135].

It is important to highlight that Section 4.2, Section 4.2.1 and Section 4.2.2 summarize the successful use of phages to control zoonotic pathogens in animals and foods, such as *Listeria monocytogenes*, *Escherichia coli*, *Campylobacter*, and *Salmonella* spp. According to the WHO, a zoonotic disease is a disease or infection that can be transmitted naturally from vertebrate animals to humans or from humans to vertebrate animals. The transmission of zoonotic pathogens impacts public health directly since it causes foodborne diseases and outbreaks [136,137].

Due to the increasing prevalence of bacterial resistance to antibiotics, new alternatives must be developed. In this scenario, the bacteriophages arise as promising alternatives to antibiotics and to control zoonotic pathogens in animals and food, as presented in the previous examples and other studies around the world [137,138,139,140].

#### 4.2.3. Biofilm Disaggregation

Biofilms are microbial communities capable of producing an extracellular polymeric matrix (EPS) of different compositions, such as polysaccharides, proteins, nucleic acids, and lipids [141]. Extracellular polymeric matrices are responsible for the three-dimensional shape of the biofilm. They are joined together and adhered to a hard surface, such as food industry equipment and product storage structures, or adhered to biological surfaces such as meat, vegetables, fruits, and other foods [142,143,144].

Using the polymeric matrices present in the biofilm, the microorganisms use dissolved nutrients and particulates, in their aqueous form, present in the medium as energy sources. In addition, the matrix can protect microorganisms against dehydration, oxidation, some antibiotics, ultraviolet radiation, and other agents [143].

Despite the numerous biotechnological applications attributed to biofilms, such as filtering drinking water, wastewater and solid waste degradation, the production of chemicals, and biofuels [145], they also raise concerns in the medical and industrial areas, as well as in the environmental area. In the health field, biofilms are associated with the contamination of medical devices and implants, for internal and external use, which culminates in several infectious diseases and great concern for human health [146,147]. In the food industry, biofilms are related to losses in food quality, quantity, and safety, given the persistence of some foodborne pathogens on food contact surfaces and biofilms [144].

In this context, bacteriophages emerge as an alternative for disaggregating biofilms and a possible solution to the challenges involving biofilms mentioned above. Phages can disaggregate biofilms through: (1) replication within their host, consequent bacterial cell lysis, and progeny release (amplification), leading to an increase in the number of phages and progressive removal of the biofilm [148]; (2) encoding enzymes—bacteriophages carry enzymes, called depolymerases, which are responsible for the degradation of EPS matrix components and the penetration of phages into the biofilm [149]; (3) the EPS matrix can also suffer destabilization due to phage infection, which consequently facilitates phage penetration and propagation within the biofilm [150]; and (4) persistent bacterial cells can also be infected by phages. Although the phage cannot replicate and destroy persistent cells, they can remain inside the bacterium until it reactivates and then destroys it [148].

##### Enzybiotics

In addition to the strategies mentioned above, other techniques do not use the phages themselves but the enzymes encoded by them. In this case, endolysins, hydrolase enzymes capable of breaking the peptidoglycan bonds present in the bacterial cell wall, can be used, triggering the release of newly formed viral particles and leading to the degradation of the bacteria and the biofilm structure [151,152,153]. According to their enzymatic activity, that is, depending on the type of peptidoglycan bond they can break, endolysins can be classified into different groups. Lytic transglycosylases (e.g., T7 phage gp16 protein) break glycosidic bonds at complex sugars, while endopeptidases (e.g., Tal2009 from Tuc2009 phage) break peptide bonds at non-terminal amino acids [154]. Lysozymes (e.g., the phage T4 gp5 protein) degrade the peptidoglycan layer by fusing with the outer membrane of the host. After degradation, the nucleocapsid crosses the host cytoplasm and drives the processes that will lead to cell lysis [155,156,157,158]. Finally, glucosaminidases are core-specific enzymes that cleave the GlcNAc-(β-1,4)-MurNAc bond. This domain is found repeatedly in *Siphoviridae* phage lytic cassettes that primarily target hosts within the *Firmicutes* family (e.g., the LambdaSa2 phage) [159,160,161,162].

Another viable alternative is the use of enzymes that are also derived from phages and are capable of degrading the extracellular substances of encapsulated bacteria, such as *Escherichia coli* K1, *Vibrio cholerae* O139, and *Pseudomonas* strains [149,163,164,165,166], facilitating the early stages of infection as well as helping to bind phages to their receptors [149,167,168,169,170].

Although progress has been made in the use of phages and their proteins to eliminate biofilms, some studies showed that they may not be sufficient to eliminate biofilms effectively. In a study, the authors applied phage to eradicate the crystalline biofilm of *Proteus mirabilis* on urinary catheters after 10 hours of infection. The results showed that the biofilm was reduced; however, the number of resident planktonic cells available to secrete biofilm was not reduced [171].

The combination of phages and antibiotics can also be applied, and some researchers already do that successfully. For example, the treatment of a *Pseudomonas aeruginosa* biofilm with a phage and ciprofloxacin decreased the biofilm significantly (22.24 to 31.07 μm^3^/μm^2^ treated with phages twice and thrice and 0.14 μm^3^/μm^2^ after ciprofloxacin) [172].

#### 4.2.4. Detection of Bacteria as Biosensor

Although considered the gold standard for bacterial detection due to their high sensitivity, culture-based methods have the disadvantage of the long period used to perform microbiological detection assays. Therefore, there is relevance in the study of new techniques capable of overcoming such disadvantages, thus aiming to reduce testing time and cost while maintaining sensitivity, specificity, and reliability during their different applications. In this sense, new techniques for detecting bacteria have been used, such as the polymerase chain reaction (PCR), immunity-based sensors (such as the enzyme-linked immunosorbent assay-ELISA), and mass spectrometry sensors [173,174].

Biosensors are structures with a component for biological recognition, a transducer, and an electronic system that amplifies, processes, and displays the signal, showing a promising alternative in bacterial detection. The most commonly used receptors are antibodies, enzymes, and nucleic acids. However, bacteriophages appear as an interesting alternative in the field of rapid detection of bacteria, as they present specific and efficient mechanisms to bind to bacteria in different environments [175].

In this topic, lytic phage-based bioassays, reporter phage systems, and phage component assays will be addressed. For detection based on a lysis assay, the phage binds to the host cell, characterizing the initial phase of its cycle, which will eventually lead to bacterial lysis. At the end of this infection cycle, phages are released from the progeny and contents cell.

Released phages can be detected by different assays, such as immunoassays, plating, or molecular approaches, such as isothermal nucleic acid amplification, mass spectrometry, enzyme immunoassay (ELISA), and quantitative PCR, the most commonly used being phage amplification and detection of progeny virions using the PCR technique [55,173,175].

Garrido and collaborators developed an amplification method for the bacteriophage *Salmonella* vB_SenS_PVP-SE2 associated with real-time PCR (qPCR) for the rapid detection of viable *Salmonella* Enteritidis in chicken samples and arrived at the detection of 8 CFU of *Salmonella* Enteritidis in 25 g of chicken samples in 10 h [176]. Anany et al. developed a phage-based paper-stick biosensor to detect various foodborne pathogens in food matrices, along with real-time quantitative PCR, which reached a threshold of 10 to 50 CFU/mL in multiple samples within 8 h of the assay [177].

From several phage amplification assays, some commercial diagnostic kits were developed for *Staphylococcus aureus*, *Yersinia pestis,* and *Mycobacterium tuberculosis*. Additionally, among the cellular components released after bacterial lysis are β-galactosidase, adenosine triphosphate (ATP), and adenylate kinase, which can be used for the target bacteria detection [175]. For example, one protocol evaluated the release of β-galactosidase through luminescent and chromogenic substrates through bacterial lysis. In this case, a LOD of 40 CFU/mL was recorded in an 8-h assay of *Escherichia coli* in water [178].

Another alternative for bacterial detection is the reporter phage systems, which are phages designed for exogenous gene insertion. These genes are inserted into bacteria and expressed by the target host. This is signaled by a fluorescent, colorimetric, electrical, or luminescent marker identifying the microorganism. In this way, phages become sensors and signal providers [175].

There are some prerequisites for the designed phage to be considered effective: (1) The virus must be able to express or translate the reporter gene in the host. (2) Identify the phage genomic region that is capable of integrating the exogenous gene. (3) Choosing a suitable detection method and the reporter gene, depending on the target, environment, and expected bacterial content. (4) Adjust the expression of the reporter gene, ensuring that production is sufficient and subsequent signal generation occurs for detection [179].

The phages chosen can have a lytic or lysogenic cycle. The systems can use different transducers to generate a signal, such as bioluminescence, colorimetry, electrochemical sensors, and fluorescence [179]. Such genes are used as reporters in Gram-positive and negative pathogen detection. However, some disadvantages prevent or lead to a lesser application of reporter phages, such as the presence of background noise caused by samples, DNA restriction-modification systems, the existence of particular genes inhibiting phages, and the antiviral activity of the defense system [173,180].

Another method for bacterial detection is not based on whole phages but on phage proteins. Such proteins include the receptor binding proteins (RBPs) and the cell-binding domains (CBDs, also called cell wall binding domains) of phage endolysins. RBPs assist in the capture and infection of the target bacteria. To identify cell capture, it is necessary to generate a signal, which is generated by the transducer, through flow cytometry, ammeter, bioluminescence, or fluorescence [173,175].

Poshtiban et al. anchored the phage NCTC12673 with RBP Gp047 on magnetic beads that were used for *Campylobacter jejuni* extraction from artificially contaminated milk and chicken broth samples. In real-time PCR results, more than 80% of the samples were infected with 10² CFU/mL of *Campylobacter* cells. By combining the magnetic separation and real-time PCR techniques, the sensitivity of the second improvement allowed the detection of *Campylobacter* cells in the samples without the need for pre-enrichment, totaling a 3-h assay [181].

Assays based on phage components are advantageous in cases where bacterial cells have not lysed or released products. Moreover, another advantage of this assay is that it uses small parts of the phage particle instead of applying the whole particle.

### 4.3. Bioengineered Phages

This topic will focus on the application of biomodified phages with a focus on phage therapy, that is, the treatment of bacterial infections, whether in animals, humans, or the environment, as there is a wide range of biotechnological applications for these viruses, which can also benefit from bioengineering (Figure 2).

Regarding the treatment of bacterial infectious diseases, the isolation of specific phages for such purposes can be difficult. It can be difficult due to a limited host range, bacterial resistance to the isolated phage, a slow replication rate, or the isolated virus may possess undesirable features such as virulence genes, antibiotic resistance genes, or lysogenic genes [182]. Therefore, genetic engineering tools can be applied in those cases where it was not possible to carry out the isolation of the most suitable phage.

Different processes are used in bioengineering for the genetic modification of bacteriophages. Homologous recombination (HR) is one of these processes. There are several techniques of traditional homologous recombination based on nucleotide sequences exchanged between different DNA molecules. In practice, HR is conducted by making a coinfection of a host cell with different phages that share similar sequences, and the progeny will result in recombinant phages. Another HR technique is phage recombination with a plasmid coding for the desirable feature, which can be designed to generate deletions, insertions, or replacements [183].

Bacteriophage Recombineering of Electroporated DNA (BRED) is another process used for phage genetic modification. It uses the recombination systems of natural phages to increase homologous recombination frequency. Electroporation of phage DNA template and donor DNA in cells expressing proteins of the phage-encoded recombination system is required to promote homologous recombination. This method can be exploited to create deletions, replacements, and heterologous gene insertions [184,185].

Phage engineering can also be based on the Clustered Regularly Interspaced Short Palindromic Repeats (CRISPR)-Cas system. CRISPR is an immune process of prokaryotic organisms. It confers immunity for the bacteria to exogenous elements and has recently been applied for bioengineering. The system acts through two main elements: Cas proteins and CRISPR RNA (crRNA). Cas enzymes cut foreign DNA into smaller fragments, mediated by crRNA. Then, the fragments (spacers) are inserted at the CRISPR locus of the bacterial genome between the repeated sequences. These changes are passed on to bacterial progenies, thus allowing them to have some immunity against infections caused by these elements.

The CRISPR-Cas system can be divided into two classes and six types. Class 1 encompasses types I, III, and IV, and class 2 is composed of types II, V, and VI [183,186]. Several studies have already demonstrated the effectiveness of using this process as a way to make changes in the genomes of different phages [187,188,189,190].

Phage genomic engineering can also be performed in vivo or in vitro. The genome is assembled and inserted into a host cell to transform it, and the phage’s replicative cycle within the bacterium will promote the assembly of new, modified viruses [183]. Genome assembly to produce the desired changes can be performed in several ways, depending on the size of the virus genome to be modified. Smaller genomes can be assembled in vitro by polymerase cycling assembly with synthetic oligonucleotides [191]. Larger genomes can be broken into smaller parts with restriction enzymes, after which these smaller genome parts can be assembled in vitro [192] or in vivo through transformation-associated recombination of overlapping genome fragments in a yeast-based assembly [193].

Recently, a three-phage cocktail with bioengineered bacteriophages was applied for the treatment of a disseminated infection caused by *Mycobacterium abscessus* in a 15-year-old patient, and significant improvement in the patient was reported after the treatment. BRED was applied to one of the phages to remove an immunity repressor gene. The authors point out that there cannot be a direct correlation between the patient’s improvement and phage therapy. However, she did not show improvement until then. In addition, patients with these conditions usually have high morbidity and mortality. This was the first therapeutic use of phages for a human mycobacterial infection as well as the first use of bioengineered phages [194].

Biofilms tend to be of great interest because they cause several problems as they help in the resistance of bacterial communities, which may be necessary for the pathogenesis of several bacteria, so it is necessary to study the interaction of phages and their enzymes with these structures [148]. Lu and Collins (2007) insert a glycoside hydrolase gene, capable of degrading biofilms, into the T7 phage that infects *Escherichia coli*. The engineered phage proved to be more efficient than the nonenzymatic parental phage, removing >99% of bacterial biofilm cells [195].

Phage bioengineering can bring several other benefits to their microbicidal potential. In addition to the examples mentioned above, HR was used to increase the host range of the T2 phage against *Escherichia coli* by introducing tail fiber genes from another phage into the T2 genome. Phages can also be modified to work as gene vectors, as demonstrated by Edgar et al. (2012), who managed to reverse antibiotic resistance through the delivery of antibiotic sensitivity genes via lysogenic phages, among an infinity of other possibilities [66,196].

It is known that phages represent the greatest diversity in the biosphere. On the other hand, little is known about the genetic diversity of these infectious entities [197,198]. To improve the ability to modify these viruses for each purpose, it is necessarily better to understand their genomic diversity and to know how to fully exploit them.

Genetically engineered phages are not readily accepted for phage therapy due to the inherent ethical issues of genetically modified organisms (GMOs), such as: (i) social concerns: access to this new technology and personal, social, and cultural consequences; (ii) extrinsic concerns: health risks associated with long-term effects on health and the environment; and (iii) intrinsic concerns: fundamental issues with creating new species [199].

### 4.4. Phage Display

Although phages were discovered a long time ago, interest in these organisms has only grown in the last few years, in line with the growing number of antibiotic-resistant bacteria and the need for new strategies to solve this problem [200].

Currently, phages have a variety of applications, such as antiviral and anticancer research, biosensor design, targeted gene or drug delivery, nanotechnology, biocontrol and detection of pathogens, and many others [128,201,202,203,204,205].

Chemical modification is a strategy that can be used as a phage display. In this approach, specific molecules of interest, such as antibiotics, anti-cancer drugs, toxins, cytokines, and others [206], are fixed on the bacteriophage surface through the chemical modification of chemical groups, including amino, acid carboxylic groups, and phenol [207]. They are incorporated both outside and inside the capsid protein envelope, allowing the use of viral particles as carriers [208].

Another approach allows the incorporation of exogenous peptides or proteins on the phage surface without any changes in the phage genome since the modifications are present in the host bacteria. Thus, when a phage infects a bacterium, wild proteins and altered proteins are randomly incorporated into the phage surface proteins and are later selected as in the classic display [209,210].

#### Phage Development for Vaccines

In this topic, phages as vaccine delivery vehicles will be addressed since the intrinsic characteristics of these organisms make them good candidates for this function.

The phage characteristics that stand out for the development of vaccines include the ease of production in bacterial host organisms on a large scale, the ease of carrying out genetic modifications, high stability in adverse environments and conditions, stability and immunogenicity of the displayed antigens, and also the ability to stimulate cellular and humoral immunity [211,212,213,214].

Phages can be applied to vaccine development in two ways: phage DNA and phage display vaccines. Phage DNA vaccines use phages to deliver DNA vaccines by incorporating the expression cassette of protective antigens or mimicked epitopes within the phage genome. DNA vaccines are stable for administration, storage, and transport. In addition, they can be orally administered [215]. However, the lead contribution of phages and the best-known system is the phage display vaccine, discussed below.

The phage display can identify and select functional polypeptides or proteins with desired immunogenetic characteristics [216]. This technology uses exogenous DNA inserted into a specific site in the nucleotide sequence that encodes a phage coat protein. Then, when the phage infects a host bacterium, the exogenous DNA is expressed as part of the coat protein along with the phage genes, and the fused protein is displayed on the exposed surface of the phage [217].

The phage display can create large peptide libraries [218]. These libraries can be natural peptide libraries (NPLs) or random peptide libraries (RPLs). The NPLs are derived from random DNA fragments from the organism of interest and incorporated into the phage genome. In contrast, RPLs are constructed by cloning synthetic random degenerate oligonucleotides and inserting them into the phage genome [217].

The generation of a phage-displayed random peptide library allows, through biopanning, the isolation of peptides that will bind to target molecules by binding affinity. The biopanning strategy consists of: (i) immobilizing the target (e.g., antibody for antigen identification) on a solid support [219], solution phase [220], or magnetic beads via organic phase separation [221]; (ii) incubating a population of phage particles with a random peptide or protein for some time, allowing the interaction between the phage-displayed peptide/protein and the target; (iii) a series of washes to remove unbound phages; (iv) elution of the phages that have maintained binding using strategies capable of breaking the bindings but maintaining phage infectivity; and (v) amplification of the eluated phages through bacterial infection. Typically, three to five rounds of affinity selection are required to reach the most strongly binding ligands, followed by isolation, enrichment, and characterization by DNA sequencing [215,222,223].

The biopanning strategy can also be performed in vivo. In this method, similar to in vitro biopanning, a random peptide library containing millions of phage clones is injected intravenously into a mouse, allowing circulation and binding to target tissues or cells. After the incubation period, the animal is sacrificed, and the desired organs or tissues are removed and eluted in saline. Washes to remove unbound, weak, and nonspecific bounds are usually performed via perfusion through the right ventricle. The tightly bound organ or tissue is amplified by infection in the host bacteria and then incubated again in another animal. After several rounds, phages displaying peptide affinity binding to the target are purified, followed by DNA sequencing to determine peptide sequences [215,217].

Filamentous phages are widely used in phage display technology, such as M13 [224,225,226], since they are particles that can be easily manipulated and are efficient in the technology of phage displays.

## 5. Final Considerations

The broad biotechnological applications presented so far, such as biopreservation and food safety, diagnostics and therapeutics, bacterial biosensors, gene transfer methods, antimicrobial therapy, vaccine carriers, DNA delivery vehicles, delivery of genes, bacteriophage display, and biofilm control, demonstrate the great potential of bacteriophages in several areas related to health, environment, and animals.

The limitations faced in phage use are also being overcome thanks to the continuous research conducted in the area, such as the expansion of the phage host range to facilitate phage therapy and interrupt the immunodominant capsid epitope to eliminate the immune response against fire and thus generate precise variants against infectious diseases [169,227,228,229]. In addition to the encapsulation technique to protect phages from the adverse conditions faced during food processing [230], there are protocols capable of expanding the host range of a phage without adding new genetic information via recombination through directed evolution in vitro [231].

In summary, the biotechnological potential of phages is an area of research that has attracted intense interest from researchers and has great potential utility. Phage research is still in its infancy, as most naturally occurring phages are not yet known, characterized, or propagated in the laboratory. Thus, further research is of utmost importance to deepen the current knowledge about these metabolically inert protein entities and to fully understand and benefit from phage biology and biotechnology.

## Figures and Tables

**Figure 1 viruses-15-00349-f001:**
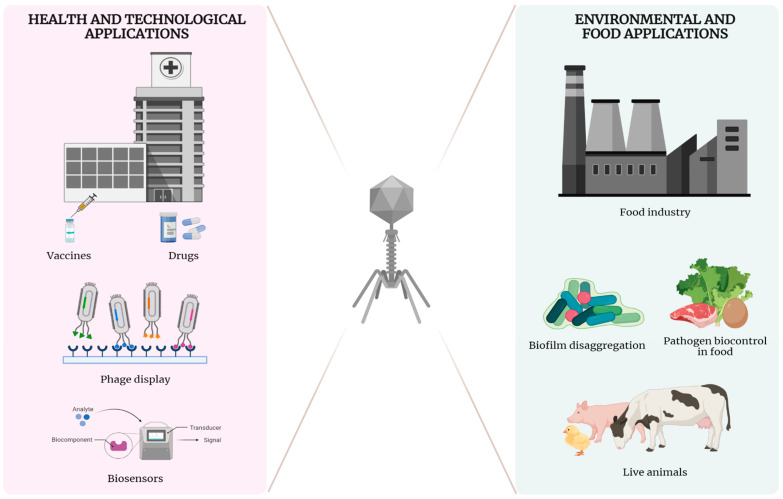
Biotechnological applications connections of bacteriophages in human, animal, and environmental areas.

**Figure 2 viruses-15-00349-f002:**
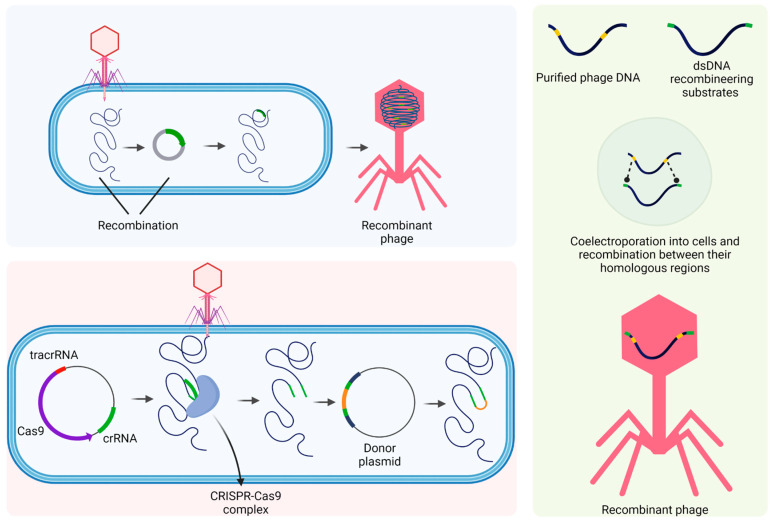
Three mechanisms for phage recombination. Homologous recombination between the wild-type phage genome and the plasmid generating phage mutants. The CRISPR-Cas9 complex, formed by tracrRNA, Cas9, and crRNA, specifically binds to the target site in the phage genome and creates a break during phage infection. Finally, the mutation was introduced into the donor plasmid, and the DNA break can be repaired by recombination with the donor to generate mutants of interest. Bacteriophage Recombineering of Electroporated DNA (BRED) consists of electroporation of the phage DNA template and donor DNA with the desired mutations flanked by homologous sequences of phage to be engineered into bacterial cells expressing proteins via either plasmid or chromosomally inserted genes to promote homologous recombination.

## Data Availability

Not applicable.

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
