# Peer review of "Bacteriophages as Biotechnological Tools"

_viruses, 2023, doi:10.3390/v15020349_

Round 1

Reviewer 1 Report

The review aims to review the biotechnological application of bacteriophages. It would like to cover a number of applications, but for me less would have been more. The title must be shortened, host range is not a central topic, applications are included in the term of of tools.

The major problem with the manuscirpt is that it is not well-organized, it was not systematically built up.

·        The morphology based taxonomy of the phages is not discussed.

·        Related to this the genomes of phages are not discussed as a potential bases of novel classification.

·        The life cycles of phages are partially discusses (2. section), it did not present a full picture. I recommend Dennehy, J.J., Abedon, S.T. (2020). Adsorption: Phage Acquisition of Bacteria. In: Harper, D., Abedon, S., Burrowes, B., McConville, M. (eds) Bacteriophages. Springer, Cham. https://doi.org/10.1007/978-3-319-40598-8_2-1

·        There is no discussion in a separate section on the role of transduction (specialized and general) in the HGT. It was mentioned that ARG can be transferred, which is a really negative process.

·        The sentences from line 109 must be explained. How, does reverse transcriptase come into the picture?

·        From line 119, genetically engineered phages are discussed – superficially.

·        The discussion of the the advantages and disadvantages of host range should be interpreted.

·        Section 4 It is a centre of the review. Fig. 1. The areas of applications are overlapping.

·        The application of phages as phage therapy killing the pathogenic bacteria has a number of criteria, advantages and limitations. These are not discussed in detail.  E.g. no effect on eukaryotic cells, immune systems are mentioned. Line 232- it is senseless, if the Authors define the only obligately lytic non-transducing phages can be used.

·        The effect of phage coctails must be explained more carefully.

·        The potential of phage therapy as a solution for zoonotic infection must be emphasized.

·        The phage based enzybiotics needs a separate section, the requirements for enzybiotics in addition to phage therapy must be presented in the case when zero tolerance had been established.

·        The biofilme disaggregation is not so simple as presented. There are many phages inable to infect bacteria in biofilms, Line 332  The sentence is strange in this form.

·        The biosensor story is also strange: line 332-333 However, all techniques 382 have the same disadvantages: specific equipment, training, and high cost. But all of the techniques presented requires similar equipments and represent more time consuming assays as compared to a simple PCR.

·        The ethical questions of GMO phages must be presented.

·        Phage display techiques should be discussed in a single paragraph. From line 569 it should be moved up.

Author Response

Dear Reviewer,

The authors are grateful for all suggestions.
We send the response below, as well as the manuscript with the requested changes.

All requested changes were added in the manuscript.

#1 - Comments and Suggestions for Authors

The review aims to review the biotechnological application of bacteriophages. It would like to cover a number of applications, but for me less would have been more. The title must be shortened, host range is not a central topic, applications are included in the term of tools.

Answer: The title was revised, maintaining the central topic of the manuscript and avoiding word redundancy.

The major problem with the manuscript is that it is not well-organized, it was not systematically built up.

Answer: To improve this aspect, requested requests were included considering the structure of the text that addresses the general aspects of bacteriophages, their uses, potential applications, and challenges. All modifications were highlighted in the manuscript.

  • The morphology-based taxonomy of the phages is not discussed.

Answer: The suggestion was accepted. It was added a paragraph about the morphology-based taxonomy of the phages.

L 42-66: “For a long time, the taxonomic classification of bacteriophages was based on their genome type, morphology, and host range. In 1937, Sir Macfarlane Burnet showed that phages are different in size and resistance against physicochemical agents [9]. In addition, Holmes Ruska proved that phages were morphologically diverse and proposed a classification of viruses by electron microscopy in 1943 [10]. The Holmes classification constituted the phages into three families based on the host range and symptoms of the disease  [11].

In 1962, Lwoff, Horne, and Tournier proposed virus classification based on the virion, its nucleic acid, and with a Latinised nomenclature that included several phages [12]. 

In 1967, David Bradley used electron microscopy and acridine orange to classify tailed bacteriophages into phages with contractile tails, long noncontractile tails, and short noncontractile tails. In 1971, this system was officially adopted by the International Committee on Nomenclature of Viruses (ICNV) and enhanced with the names Myoviridae, Styloviridae, and Pedoviridae for the three morphotypes by Hans-Wolfgang Ackermann & Abraham Eisenstark of the Bacterial Virus Subcommittee in 1975 [13–16].

The family phages' names Myoviridae and Podoviridae, as known currently, were accepted by the International Committee on Taxonomy of Viruses (ICTV) in 1981 and Siphoviridae in 1984. Following the phage taxonomy story, the order Caudovirales (comprising Myoviridae, Siphoviridae, Podoviridae, Ackermannviridae, and Herelleviridae families), unifying all tailed phages, was proposed by Hans-Wolfgang Ackermann and approved in 1988 [17].

In the same way, based on the morphology, others phages families were classified such as Inoviridae (filamentous capsid), Microviridae, Tectiviridae, Corticoviridae, Leviviridae, and Cystoviridae (polyhedral capsid), and Plasmaviridae (pleomorphic capsid) in 1978 [17].”

  • Related to this the genomes of phages are not discussed as a potential bases of novel classification.

Answer: The suggestion was accepted. It was added a paragraph about the genomes of phages as potential bases of novel classification.

L 67-73: “However, with the advent of genomics, the sequencing of phage genomes revealed a higher genomic diversity from the human gut to the deep ocean [18]. Besides, a genome organization-based taxonomy included new subfamilies and genera and virus relatedness based upon nucleotide or more commonly protein sequence or proteome comparisons. Based on that, molecular analysis can provide a robust classification guide for phages, even for those with highly divergent genome sequences and organizations, high rates of horizontal gene transfer, and mosaicism [19–23].”

  • The life cycles of phages are partially discusses (2. section), it did not present a full picture. I recommend Dennehy, J.J., Abedon, S.T. (2020). Adsorption: Phage Acquisition of Bacteria. In: Harper, D., Abedon, S., Burrowes, B., McConville, M. (eds) Bacteriophages. Springer, Cham. https://doi.org/10.1007/978-3-319-40598-8_2-1

Answer: The suggestion was accepted. The life cycles of the bacteriophages were presented in more detail.

L 96-125: “Bacteriophages are classified as lytic and/or lysogenic according to their replication cycle. The first step of the lytic cycle consists of the virion's encounter with the bacteria, a process also called adsorption. In this step, the virion movement toward the bacteria can be differentiated into diffusion, bulk, relative, or turbulent movement [27]. The following step consists of the phage's reversible attachment to the host bacteria. The attachment is firstly reversible because there are no permanent changes in virion morphology. Then, the irreversible attachment occurs thanks to the binding of the secondary attachment proteins to the secondary receptor, which is stronger than the binding of the primary attachment protein to the primary receptor in the reversible attachment [27]. The next step of the lytic cycle consists of the virion genome translocation to the bacteria cytoplasm. To translocate their genomes, phages have to surpass the bacterial surface structures (e.g., glycocalyx, S-layers, and peptidoglycan cell walls), for that, most phages use their enzymes targeting these structures and introduce its nucleic acid into the host cytoplasm via a combination of mechanical and enzymatic action that results in a process known as genome injection. [28–30]

Into the bacterial cytoplasm, the phage genes are expressed and its genome is replicated, starting the virion morphogenesis. The next step comprises a phage virion's accumulation intracellularly, at a constant rate, until phage-induced bacterial lysis. In some cases, loss of infection viability or cessation of phage production for chronically infecting phages can occur [31].

The lytic cycle of phages is of great interest for the biological control of bacteria, as they can lysate the host cell, and the resulting progeny continue the lytic cycle.

In the lysogenic cycle, the movement, attachment, and infection of the phage to the host bacteria occur similarly to the lytic cycle. However, in the lysogenic cycle, the phage's nucleic acid recombines with the bacteria's nucleic acid, forming a prophage, which replicates with the host chromosome and is transferred vertically from the initially infected cell to its progeny through cell division. The lysogenic cycle may provide immunity against infection by the same type of phage. Moreover, stress conditions such as ultraviolet light or mutagenic chemicals, or DNA damage can induce a shift to the lytic cycle [32,33]”.

  • There is no discussion in a separate section on the role of transduction (specialized and general) in the HGT. It was mentioned that ARG can be transferred, which is a really negative process.

Answer: The suggestion was accepted. It was added a separate section on the role of transduction (specialized and general) in the HGT. The sentence mentioned that ARG can be transferred was removed.

L 323-344: “4.1.1. Horizontal gene transfer (HGT) in prokaryotes

There are three mechanisms of horizontal gene transfer (HGT) in prokaryotes, called transformation, conjugation, and transduction [98]. Bacteriophages mediate HGT through transduction where the DNA from a donor bacterial cell is transferred to a recipient bacterial cell. Transduction can be specialized or generalized. The first one can occur only with temperate phages and is generated from the imprecise excise of adjacent host genes together with the phage genome. Therefore, the host DNA becomes part of the phage genome, is replicated, and all the virions produced by the cell will carry the fragment of the host DNA [33]

Bacteriophage lambda is a classic example of specialized transducing phage since its integration into the genome of Escherichia coli, transfers genes related to the galactose metabolism and biotin biosynthesis, and confers a fitness advantage in certain environments.

The generalized transduction in turn can occur by both lytic or lysogenic cycle and consists of random parts of the host DNA packed in bacteriophages. Briefly, the terminase complex recognizes a pac site homolog in the host chromosome and introduces a double-stranded break nearby. When the small subunit of the terminase complex recognizes the pac sequence, the large subunit starts to package the DNA into the procapsid until the capacity of the phage head is reached and a second double-stranded break is introduced. The filled virion passes through the tail maturation and detaches from the terminase complex. Then, the terminase complex attaches to the free end of the bacterial chromosome and encapsidate phage-sized fragments of the host genome until it disengages from the chromosome [33]”.

  • The sentences from line 109 must be explained. How, does reverse transcriptase come into the picture?

Answer: The paragraph about the reverse transcriptase was revised and written with more details, as follows: L 163-169: “Bacteriophages can alter their host range by modifications in the RBPs. Some phages can alter their RBPs over the generation with a mechanism based on the enzyme reverse transcriptase. The temperate Bordetella phage BPP-1 is an example of a phage that generates diversity in a gene and specifies different tropisms for receptor molecules by reverse transcriptase–mediated process. By the activity of this enzyme, occurs nucleotide substitutions that result in tropism switching. And based on that, a huge repertoire of ligand-receptor interactions is generated”.

  • From line 119, genetically engineered phages are discussed – superficially.

Answer: We appreciate the reviewer’s contribution. The genetically engineered phages are discussed in more detail in section 4.3.

  • The discussion of the advantages and disadvantages of host range should be interpreted.

Answer: To clarify this point, the following paragraph has been improved:

L 163-168: Bacteriophages can alter their host range by modifications in the RBPs. Some phages can alter their RBPs over the generation with a mechanism based on the enzyme reverse transcriptase. The temperate Bordetella phage BPP-1 is an example of a phage that generates diversity in a gene and specifies different tropisms for receptor molecules by reverse transcriptase–mediated process. By the activity of this enzyme, occurs nucleotide substitutions that result in tropism switching. And based on that, a huge repertoire of ligand-receptor interactions is generated [43].

  • Section 4 It is a center of the review. Fig. 1. The areas of applications are overlapping. 

Answer: The legend has been improved and shows the interconnections of phage uses in human, animal, and environmental areas. “Biotechnological applications connections of bacteriophages in human, animal, and environmental areas.”

The application of phages as phage therapy killing the pathogenic bacteria has a number of criteria, advantages and limitations. These are not discussed in detail.  E.g. no effect on eukaryotic cells, immune systems are mentioned. Line 232- it is senseless, if the Authors define the only obligately lytic non-transducing phages can be used.

Answer: The section on phage therapy was reformulated, adding more details about the advantages and limitations of the method. The sentence in line 232 was removed.

L 547-256: “4.1. Human Phage Therapy: advantages and limitations

Phage therapy is a method to combat bacterial infections using bacteriophages and has been a major focus of attention in recent years. This approach mainly uses viruses that perform an obligatory lytic cycle, so that it kills the pathogen without necessarily promoting any changes in its genome. The advantages of bacteriophage use include: i) its specificity for target bacteria in event of clinical application that may considerably reduce the damage to the patient’s intestinal microbiota; ii) self-limiting growth, meaning that they require their hosts to be constantly growing; iii) if the bacterial pathogens for which they are specific are absent, they won't persist long enough, and lastly; iv) replication at the site of infection”.

L 275-303: “For a long time, it was thought that the bacteriophages couldn't affect eukaryotic cells. However, recent studies have provided new insights about the interactions between them.

Bacteriophages can interact with the surfaces of mucus, for example. An in vitro experiment using a T4 bacteriophage and mucus-producing tissue cell line showed a binding between the bacteriophages and glycan residues displayed on mucin glycoproteins via Ig-like domains present in Hoc capsid protein [75]. Since the attached bacteriophages have a higher probability to encounter the host bacteria than free virions, it is proposed that mucus adherence may help shape the gastrointestinal microbiome and prevent pathogens from colonizing the system [75–77].

The bacteriophages may also interact with cells of the immune system. A study assessed the effects of the bacteriophage ES2 on the expression of surface proteins CD86, CD40, and MHCII, the production of pro-inflammatory cytokines IL-6, IL-1α, IL-1β, and TNF-α by dendritic cells, and the activation of the NF-κB signaling pathway. The results showed that ES2 increased the expression of surface proteins and pro-inflammatory cytokines. Besides, the study also showed the activation and translocation of NF-κBp65 to the nucleus, leading to the activation of NF-κB signaling [78]. In the same way, another study observed the activation status of mammalian macrophages and TNF-α levels by two bacteriophages of Escherichia coli.

Cells of the immune system, such as cited above, express pattern recognition receptors. These receptors include a family of receptors called Toll-like receptors that are capable of recognizing bacteriophage nucleic acids in endosomes.

Besides the immune system, there are studies presenting interactions of bacteriophages with the respiratory system [79,80], central nervous system [81], gastrointestinal tract [77], urinary tract [82,83], and cancer [84–86], to cite some.

Among the disadvantages of phage therapy, can be punctuated with the release of bacterial toxins (e.g., endotoxins) when the bacteria are lysed, which can worsen bacterial infection[87,88]. Besides, foreign proteins carried by bacteriophages may induce overreaction or cause an imbalance in the immune system of humans or animals [89,90].

  • The effect of phage cocktails must be explained more carefully.

Answer: The suggestion was accepted. The effect of phage cocktails was presented in more detail.

L 307-314: “Related to bacteriophage cocktails, this approach arises due to the high specificity of phage which often leads to limitations in identifying the fitting strain. In emergency cases, for example, where there is a search for a corresponding strain of phage before treatment, the use of phage cocktails can increase the efficiency of pairing by increasing the range of action [94,95]. Phage cocktails can also delay the emergence of phage-resistant bacteria since multiple phages are interacting with bacteria. In addition, different strains of phages can complement one another by providing the necessary antimicrobial elements that one may be short of [94]”.

  • The potential of phage therapy as a solution for zoonotic infection must be emphasized.

Answer: We appreciate the reviewer’s contribution. The potential of phage therapy as a solution for zoonotic infection was emphasized at the end of section 4.2.2.

L 420-430: “It is important to highlight that the sections 4.2, 4.2.1, and 4.2.2 summarizes the successful use of phages to control zoonotic pathogens in animals and foods, such as Listeria monocytogenes, Escherichia coli, Campylobacter, and Salmonella spp. According to WHO, a zoonotic disease is a disease or infection that can be transmitted naturally from vertebrate animals to humans or from humans to vertebrate animals. The transmission of zoonotic pathogens impacts public health directly since it causes foodborne diseases and outbreaks [127,128].

Due to the increasing bacterial resistance prevalence to antibiotics, new alternatives must be developed. In this scenario, the bacteriophages arise as promising alternatives to antibiotics and to control zoonotic pathogens in animals and food, as presented in the previous examples and other studies around the world [128–131]”.

  • The phage based enzybiotics needs a separate section, the requirements for enzybiotics in addition to phage therapy must be presented in the case when zero tolerance had been established.

Answer: The suggestion was accepted. It was added a separate section (4.2.3.1) for enzybiotics.

  • The biofilm disaggregation is not so simple as presented. There are many phages enable to infect bacteria in biofilms, Line 332  The sentence is strange in this form.

Answer: The suggestion was accepted. The biofilm disaggregation text was reformulated.

L 481-490: “Although progress in the use of phages and their proteins to eliminate biofilms, some studies showed that they may not be sufficient to eliminate biofilms effectively. In a study, the authors applied phage to eradicate the crystalline biofilm of Proteus mirabilis on urinary catheters after ten hours of infections. The results showed that the biofilm was reduced, however, the number of resident planktonic cells available to secrete biofilm was not [162].

The combination of phages and antibiotics can also be applied, and some researchers already do that successfully. For example, the treatment of a Pseudomonas aeruginosa biofilm with phage and ciprofloxacin decreased the biofilm significantly (22.24 to 31.07 μm3/μm2 treated with phages twice and thrice and 0.14 μm3/μm2 after ciprofloxacin) [163]”.

The sentence in line 332 was also reformulated.

L 438-441: “Using the polymeric matrices present in the biofilm, the microorganisms use dissolved nutrients and particulates, in their aqueous form, present in the medium as energy sources”.

  • The biosensor story is also strange: line 332-333 However, all techniques 382 have the same disadvantages: specific equipment, training, and high cost. But all of the techniques presented requires similar equipment and represent more time-consuming assays as compared to a simple PCR.

Answer: We appreciate the reviewer’s contribution. The sentence was reformulated.

L 492-499: “Although considered the gold standard for bacterial detection due to its high sensitivity, culture-based methods have the disadvantage of the long period used to perform microbiological detection assays. Therefore, there is relevance in the study of new techniques capable of overcoming such disadvantages, thus aiming at reducing testing time and cost, as well as maintaining sensitivity, specificity and reliability during their different applications. In this sense, new techniques for detecting bacteria have been used, such as the polymerase chain reaction (PCR); immunity-based sensors (such as the enzyme-linked immunosorbent assay-ELISA), and mass spectrometry sensors [164,165]”.

  • The ethical questions of GMO phages must be presented.

Answer: We appreciate the reviewer’s contribution. The ethical questions of GMO phages were emphasized at the end of section 4.3.

L 636-640: “Genetically engineered phages are not readily accepted for phage therapy due to the inherent ethical issues of genetically modified organisms (GMOs) such as: i) social concerns: access to this new technology, and personal, social, and cultural consequences; ii) extrinsic concerns: health risk associated, long-term effects in the health and environment; and iii) intrinsic concerns: fundamental issues with creating new species [190]”.

  • Phage display techniques should be discussed in a single paragraph. From line 569 it should be moved up.

Answer: We appreciate the reviewer’s contribution. We thought that keeping the initial text can help the reader to understand the theme. Since the main characteristics of the phage for the development of vaccines are presented, and also how the phages can be used for the vaccine.

Reviewer 2 Report

Elois et al have working on summarizing the application of bacteriophages as a tool in biotechnology. They did a great job of going over the mechanism of infection of these phage. Their summary on the application of bacteriophage in healthcare as antimicrobials in humans and animals, biofilm disruptors, and biosensors is quite well written. The description of bioengineered phage as well as phage display tool was helpful in increasing the impact of the review.

There are a few opportunities to improve the clarity of the manuscript.

Please ensure words like in vitro are consistently italicized.

E. coli is sometimes spelled fully vs abbreviated at times. Please ensure that it is consistent throughout the manuscript.

Page 2 Line 57 – “referrer” is a typo here.

Page 2 Line 94 – Do the authors mean “To complete the invasion” instead of “To the complete invasion”.

Page Line 152-161 – Please add more references that discuss the CRISPR mechanism in details.

Page 4 Line 169-171 – Please add references here that discuss phage in agriculture and food industry.

Page 4 Line 174 – “phage therapy” is repeated in this sentence.

Page 4 Line 175 – Please re-word as the highlighted region is the sentence doesn’t seem to convey the message properly “The bacteriophage's application for treatment depends on their host range well characterized and”

Page 5 Line 210 – Please re-phrase the sentence “There are considered a defective structure resembling the tail of bacteriophages without a head and absence of a genome, and as phages, PTLBs also depend on RBPs to 211 target a host bacterium” to convey the correct message.

Page 6 Line 234 – Do the authors mean “And they increase the dynamics of biofilm formation…” instead of “And to increase the dynamics of biofilm formation…” as the current sentence sounds incomplete.

Page 6 Line 252 – The sentence is incomplete.

Page 6 Line 253-254 – Please add references for uses of phage in food, health, and sanitary.

Page 10 Line 445 – “Also, using small parts of the particle instead of applying the whole particle.” sentence is incomplete.

Author Response

Dear Reviewer,

The authors are grateful for all suggestions from the reviewers. We send the response letter as an attachment, as well as the manuscript with the requested changes. All requested changes were added in the manuscript.

#2 - Comments and Suggestions for Authors

Elois et al have working on summarizing the application of bacteriophages as a tool in biotechnology. They did a great job of going over the mechanism of infection of these phages. Their summary on the application of bacteriophage in healthcare as antimicrobials in humans and animals, biofilm disruptors, and biosensors is quite well written. The description of bioengineered phage as well as phage display tool was helpful in increasing the impact of the review.

Answer: We appreciated the comments a lot!!!

There are a few opportunities to improve the clarity of the manuscript.

Please ensure words like in vitro are consistently italicized.

Answer: We appreciate the reviewer’s contribution. All the words in vitro were carefully revised.

  1. coli is sometimes spelled fully vs abbreviated at times. Please ensure that it is consistent throughout the manuscript.

Answer: The suggestion was accepted. All Escherichia coli words were carefully revised and spelled fully throughout the manuscript.

Page 2 Line 57 – “referrer” is a typo here.

Answer: The word was revised, as suggested.

Page 2 Line 94 – Do the authors mean “To complete the invasion” instead of “To the complete invasion”.

Answer: We appreciate the reviewer’s contribution. The sentence was altered, as suggested.

Page Line 152-161 – Please add more references that discuss the CRISPR mechanism in details.

Answer: The suggestion was accepted. The two references below were included and discuss the CRISPR mechanism in detail.

Hryhorowicz, M., LipiÅ„ski, D., Zeyland, J. et al. CRISPR/Cas9 Immune System as a Tool for Genome Engineering. Arch. Immunol. Ther. Exp. 65, 233–240 (2017). https://doi.org/10.1007/s00005-016-0427-5

Devashish Rath, Lina Amlinger, Archana Rath, Magnus Lundgren, The CRISPR-Cas immune system: Biology, mechanisms and applications, Biochimie, Volume 117, 2015, Pages 119-128, ISSN 0300-9084, https://doi.org/10.1016/j.biochi.2015.03.025.

Page 4 Line 169-171 – Please add references here that discuss phage in agriculture and food industry.

Answer: The suggestion was accepted. The references below were included and discuss phage in agriculture and the food industry.

Agriculture

Jones, H.J.; Shield, C.G.; Swift, B.M.C. The Application of Bacteriophage Diagnostics for Bacterial Pathogens in the Agricultural Supply Chain: From Farm-To-Fork. PHAGE: Therapy, Applications, and Research 2020, 1, 176–188, doi:10.1089/PHAGE.2020.0042/ASSET/IMAGES/LARGE/PHAGE.2020.0042_FIGURE1.JPEG.

Nakayinga, R.; Makumi, A.; Tumuhaise, V.; Tinzaara, W. Xanthomonas Bacteriophages: A Review of Their Biology and Biocontrol Applications in Agriculture. BMC Microbiol 2021, 21, 1–20, doi:10.1186/S12866-021-02351-7/TABLES/5.

Sasaki, R.; Miyashita, S.; Ando, S.; Ito, K.; Fukuhara, T.; Takahashi, H. Isolation and Characterization of a Novel Jumbo Phage from Leaf Litter Compost and Its Suppressive Effect on Rice Seedling Rot Diseases. Viruses 2021, 13, 591, doi:10.3390/V13040591/S1.

Food industry

Cristobal-Cueto, P.; García-Quintanilla, A.; Esteban, J.; García-Quintanilla, M. Phages in Food Industry Biocontrol and Bioremediation. Antibiotics 2021, Vol. 10, Page 786 2021, 10, 786, doi:10.3390/ANTIBIOTICS10070786.

Kawacka, I.; Olejnik-Schmidt, A.; Schmidt, M.; Sip, A. Effectiveness of Phage-Based Inhibition of Listeria Monocytogenes in Food Products and Food Processing Environments. Microorganisms 2020, Vol. 8, Page 1764 2020, 8, 1764, doi:10.3390/MICROORGANISMS8111764.

Page 4 Line 174 – “phage therapy” is repeated in this sentence.

Answer: The word was revised, as suggested.

Page 4 Line 175 – Please re-word as the highlighted region is the sentence doesn’t seem to convey the message properly “The bacteriophage's application for treatment depends on their host range well characterized and”

Answer: We appreciate the reviewer’s contribution. The sentence was reformulated as follows: L 230-232: “The use of phages for the treatment of bacterial infections is called phage therapy. And can only be applied if the host range is well characterized and limited to the pathogen of interest”.

Page 5 Line 210 – Please re-phrase the sentence “There are considered a defective structure resembling the tail of bacteriophages without a head and absence of a genome, and as phages, PTLBs also depend on RBPs to 211 target a host bacterium” to convey the correct message.

Answer: We appreciate the reviewer’s contribution. The sentence was reformulated as follows: L 267-270: “Another tool for antibacterial purposes is phage tail-like bacteriocins (PTLBs) or tailoring. These structures are resembling the tail of bacteriophages, but without a head and absence of a genome, and as phages, PTLBs also depend on RBPs to target a host bacterium”.

Page 6 Line 234 – Do the authors mean “And they increase the dynamics of biofilm formation…” instead of “And to increase the dynamics of biofilm formation…” as the current sentence sounds incomplete.

Answer: We appreciate the reviewer’s contribution. The sentence was altered, as suggested.

Page 6 Line 252 – The sentence is incomplete.

Answer: We appreciate the reviewer’s contribution. The sentence was altered, as follows: L 346-348: “The biotechnological applications achieved using bacteriophages are diverse and still growing, since the administration of phages directly to live animals until to ready-to-eat food”.

Page 6 Line 253-254 – Please add references for uses of phage in food, health, and sanitary.

Answer: The suggestion was accepted. The references below were included and discuss phage in food, health, and sanitary.

Corpuz, A. v Potential of Bacteriophage Therapy in Treating Hospital Wastewater. European Journal of Molecular & Clinical Medicine 2020, 7, 2020.

Ballesté, E.; Blanch, A.R.; Muniesa, M.; García-Aljaro, C.; Rodríguez-Rubio, L.; Martín-Díaz, J.; Pascual-Benito, M.; Jofre, J. Bacteriophages in Sewage: Abundance, Roles, and Applications. FEMS Microbes 2022, 3, doi:10.1093/FEMSMC/XTAC009.

Guo, Z.; Lin, H.; Ji, X.; Yan, G.; Lei, L.; Han, W.; Gu, J.; Huang, J. Therapeutic Applications of Lytic Phages in Human Medicine. Microb Pathog 2020, 142, 104048, doi:10.1016/J.MICPATH.2020.104048.

Qin, L.; Huiwen, M.; Wang, J.; Wang, Y.; Khan, S.A.; Zhang, Y.; Qiu, H.; Jiang, L.; He, L.; Zhang, Y.; et al. A Novel Polymerase β Inhibitor from Phage Displayed Peptide Library Augments the Anti-Tumour Effects of Temozolomide on Colorectal Cancer. https://doi.org/10.1080/1120009X.2021.2009987 2021, 34, 391–400, doi:10.1080/1120009X.2021.2009987.

Yan, T.; Liang, L.; Yin, P.; Zhou, Y.; Sharoba, A.M.; Lu, Q.; Dong, X.; Liu, K.; Connerton, I.F.; Li, J. Application of a Novel Phage LPSEYT for Biological Control of Salmonella in Foods. Microorganisms 2020, Vol. 8, Page 400 2020, 8, 400, doi:10.3390/MICROORGANISMS8030400.

Waturangi, D.E.; Kasriady, C.P.; Guntama, G.; Sahulata, A.M.; Lestari, D.; Magdalena, S. Application of Bacteriophage as Food Preservative to Control Enteropathogenic Escherichia Coli (EPEC). BMC Res Notes 2021, 14, 1–6, doi:10.1186/S13104-021-05756-9/TABLES/1.

Islam, M.S.; Yang, X.; Euler, C.W.; Han, X.; Liu, J.; Hossen, M.I.; Zhou, Y.; Li, J. Application of a Novel Phage ZPAH7 for Controlling Multidrug-Resistant Aeromonas Hydrophila on Lettuce and Reducing Biofilms. Food Control 2021, 122, 107785, doi:10.1016/J.FOODCONT.2020.107785.

Page 10 Line 445 – “Also, using small parts of the particle instead of applying the whole particle.” sentence is incomplete.

Answer: We appreciate the reviewer’s contribution. The sentence was altered, as follows: L 560-562: “Assays based on phage components are advantageous in cases where bacterial cells have not lysed or released products. Moreover, other advantageous this assay is using small parts of the phage particle instead of applying the whole particle”.

Round 2

Reviewer 1 Report

The manuscript was improved according to most of the request.

However, since it is a general review on the biotechnological applications of bacteriophages, I insist on the detailed presentation of phage life cycles. Although, I suggested a paper to help, I suggest one more: Zack Hobbs and Stephen T. Abedon Diversity of phage infection types and associated terminology: the problem with ‘Lytic or lysogenic’ FEMS Microbiology Letters, 363, 2016, fnw047, doi: 10.1093/femsle/fnw047

line  164, Please, provide a more detailed, understandable description of  the reverse transciptase based RBS modification

lines 250-258 This is not limited to Human Phage Therapy. These must go up to line 233.

Author Response

Dear Editor,

On behalf of the co-authors, I would like to submit and be considered for publication in the second revision of the manuscript Manuscript ID viruses-2083267.

The authors are grateful for all suggestions from the reviewer. We send the response letter as an attachment, as well as the manuscript with the requested changes.

#1 - Comments and Suggestions for Authors

However, since it is a general review on the biotechnological applications of bacteriophages, I insist on the detailed presentation of phage life cycles. Although I suggested a paper to help, I suggest one more: Zack Hobbs and Stephen T. Abedon Diversity of phage infection types and associated terminology: the problem with ‘Lytic or lysogenic’ FEMS Microbiology Letters, 363, 2016, fnw047, doi: 10.1093/femsle/fnw047

Answer: The suggestion was accepted. The life cycles of the bacteriophages were presented in more detail. The improved sections are highlighted in green.

L 98-166: “In terms of the characterization of phage life cycles, there are strategies for infection and release. Related to encapsidation and location, phages can be: i) intracellularly and unencapsidated. This state can be subdivided as a “Vegetative phase” as described by Lwoff, 1953, or productive cycle, versus as a prophage; ii) intracellularly and packaged within mature virions, in other words, can be distinguished from phage genomes and therefore are not packaged until the virion release step; and iii) intracellularly and encapsidated. In this case, free phages are no longer found within their bacterial host [27,28].

Based on these strategies of infection and release, phages can be lytic and non-temperate, in other words when lytic phages do not display lysogenic cycles, or chronic and non-temperate, that is when chronically released phages that do not display lysogenic cycles. In the first strategy, the phage passes through a vegetative phase, and its genome is packaged into mature virions before the release of free phages. The second case is similar to the previous one, the phage passes through a vegetative phase, however, its genome is packaged into mature virions during the release of free phages [28].

The lytic and non-temperate phages are of great interest for the biological control of bacteria, as they can lysate the host cell, and the resulting progeny continue the cycle [29].

The next two strategies of infection and release of a phage include lytic and temperate phages, in other words, lytic phages that can display lysogenic cycles, or chronic and temperate phages, that is, chronically released phages that can display lysogenic cycles. In this scenario, in the first strategy, the phage can display a vegetative or prophage phase, and its genome is packaged into mature virions before the release of free phages. And similar to this, in the second strategy the phage can display a vegetative or prophage phase, however, its genome is packaged into mature virions during the release of free phages [28].

The first step of a phage’s life cycle consists of the virion's encounter with the bacteria, a process also called adsorption. In the adsorption process, the virion movement toward the bacteria can be differentiated into diffusion and non-diffuse movement (relative, turbulent, or bulk) [30]. The diffusion movement occur when the phages are not attached or entanglement with materials. In addition, this movement can be influenced by different factors such as particle size and morphology [31,32]

The non-diffuse movement is related phage binding to nonhost materials can result in virion retention within fluids along with virion movement in association with ongoing currents. The non-diffuse movement can increase the likelihood of phage encounter with a target cell if virions are moving in such flow relative to the target bacteria. If the movement is a turbulent flow (e.g., microenvironments mixed), that can result instead in faster virion movement relative to bacteria [33]. Lastly, virions also may be transported over longer distances such as through the air, associated with dust, splashed water, or animals serving as mechanical vectors [34–36]. However, is important to note that this bulk movement is not necessarily relative to the positions of co-located bacteria. Therefore, it may not contribute to phage encounter with susceptible bacteria [30].

The following step of the adsorption process consists of the phage's reversible attachment to the host bacteria. The attachment is firstly reversible because there are no permanent changes in virion morphology. Then, the final step of the adsorption process culminates in free virion irreversible attachment to a bacterial cell. The irreversible attachment occurs thanks to the binding of the secondary attachment proteins to the secondary receptor, which is stronger than the binding of the primary attachment protein to the primary receptor in the reversible attachment [30].

The next step of a phage’s life cycle consists of phage acquisition of bacteria, that is, the conversion of a free virion (seeds or spores for multicellular organisms) to a virocell (‘living form’ of the virus) [37]. In this step, the virion genome is translocated to the bacteria cytoplasm. To translocate their genomes, phages have to surpass the bacterial surface structures (e.g., glycocalyx, S-layers, and peptidoglycan cell walls), for that, most phages use their enzymes targeting these structures and introduce its nucleic acid into the host cytoplasm via a combination of mechanical and enzymatic action that results in a process known as genome injection [38–40]

Into the bacterial cytoplasm, the phage genes are expressed and its genome is replicated, starting the virion morphogenesis. The next step comprises a phage virion's accumulation intracellularly, at a constant rate, until phage-induced bacterial lysis. In some cases, loss of infection viability or cessation of phage production for chronically infecting phages can occur as discussed before [41].

The movement, attachment, and genomes translocation of the phage to the host bacteria occur similarly to the described before with chronic and temperate phages or lytic and temperate phages displaying lysogenic cycles. However, in the lysogenic cycle, the phage's nucleic acid recombines with the bacteria's nucleic acid, forming a prophage, which replicates with the host chromosome and is transferred vertically from the initially infected cell to its progeny through cell division. The lysogenic cycle may provide immunity against infection by the same type of phage. Moreover, stress conditions such as ultraviolet light or mutagenic chemicals, or DNA damage can induce a shift to the lytic cycle [42]”.

Line  164, Please, provide a more detailed, understandable description of  the reverse transciptase based RBS modification

Answer: The suggestion was accepted. The description of the reverse transcriptase based RBP modification were presented in more detail.

L 206-225: ”The temperate Bordetella phage BPP-1 is an example of a phage that generates diversity in a gene, designated Major Tropism Determinant (MTD) , and specifies different tropisms for receptor molecules by reverse transcriptase–mediated process. The reverse transcriptase enzyme introduces nucleotide substitutions at defined locations within MTD that result in tropism switching. And based on that, a huge repertoire of ligand-receptor interactions is generated. Briefly, in this study, searching to understand the tropism presented by the BvgAS signal transduction system that controls the infectious cycles of Bordetella subspecies, the researchers found a region of variability, VR1, which differed between tropic variants. Located downstream from MTDis a second template repeat (TR) that never varied when sequences of phage with similar or different tropisms were compared. Adjacent to TR is a locus, called bordetella reverse transcriptase (BRT), which encodes an enzymatically active reverse transcriptase (RT) similar to the RT domains of group II intron maturases, bacterial retrons, and retroviral reverse transcriptases. Based on these, the researchers constructed a series of in-frame deletion and substitution mutations to determine the roles of VR1, the TR element, and the BRT locus in phage infectivity and tropism switching. Deletion mutations in the BRT loci of two variants resulted in fully infective phages that had completely lost the ability to switch tropism. Altering the conserved reverse transcriptase motif eliminated BRT activity in vitro and tropism switching in vivo. Thus, they concluded tropism switching is a reverse transcriptase–mediated event. [52]”.

lines 250-258 This is not limited to Human Phage Therapy. These must go up to line 233.

Answer: The suggestion was accepted. 

L 288-296: “The use of phages for the treatment of bacterial infections is called phage therapy. Phage therapy is a method to combat bacterial infections using bacteriophages and has been a major focus of attention in recent years. This approach mainly uses viruses that are professionally lytic cycle, so that it kills the pathogen without necessarily promoting any changes in its genome. The advantages of bacteriophage use include: i) its specificity for target bacteria in event of clinical application that may considerably reduce the damage to the patient’s intestinal microbiota; ii) self-limiting growth, meaning that they require their hosts to be constantly growing; iii) if the bacterial pathogens for which they are specific are absent, they won't persist long enough, and lastly; iv) replication at the site of infection”.